# Exploring the Epigenome in Gastroenteropancreatic Neuroendocrine Neoplasias

**DOI:** 10.3390/cancers13164181

**Published:** 2021-08-20

**Authors:** Rohini Sharma, Mark P. Lythgoe, Bhavandeep Slaich, Nishil Patel

**Affiliations:** 1Department of Surgery and Cancer, Imperial College London, London W12 ONN, UK; m.lythgoe@imperial.ac.uk; 2Department of Medicine, University of Leicester, Leicester LE1 7RH, UK; bhavandeep.slaich@nhs.net (B.S.); Nishil.patel@royalberkshire.nhs.uk (N.P.)

**Keywords:** neuroendocrine neoplasias, epigenetic, methylation, miRNA, histone

## Abstract

**Simple Summary:**

There is increasing recognition of the role of epigenetics in facilitating the pathogenesis and clinical outcome in patients with a diagnosis of neuroendocrine neoplasia. In this review we outline the different types of epigenetic changes that are observed and their impact. We also highlight novel interventions that can be used to manipulate these epigenetic changes that are being explored in the clinic.

**Abstract:**

Gastroenteropancreatic neuroendocrine neoplasias are a diverse group of neoplasms with different characteristics in terms of site, biological behaviour and metastatic potential. In comparison to other cancers, they are genetically quiet, harbouring relatively few somatic mutations. It is increasingly becoming evident that epigenetic changes are as relevant, if not more so, as somatic mutations in promoting oncogenesis. Despite significant tumour heterogeneity, it is obvious that DNA methylation, histone and chromatin modifications and microRNA expression profiles are distinctive for GEP-NEN subtypes and may correlate with clinical outcome. This review summarises existing knowledge on epigenetic changes, identifying potential contributions to pathogenesis and oncogenesis. In particular, we focus on epigenetic changes pertaining to well-differentiated neuroendocrine tumours, which make up the bulk of NENs. We also highlight both similarities and differences within the subtypes of GEP-NETs and how these relate and compare to other types of cancers. We relate epigenetic understanding to existing treatments and explore how this knowledge may be exploited in the development of novel treatment approaches, such as in theranostics and combining conventional treatment modalities. We consider potential barriers to epigenetic research in GEP-NENs and discuss strategies to optimise research and development of new therapies.

## 1. Introduction

Neuroendocrine neoplasias (NENs) are a heterogeneous group of neoplasms that arise from cells of the endocrine glands as well as the diffuse neuroendocrine system [1]. Gastroenteropancreatic NENs (GEP-NENs) comprise over two-thirds of all NENs and may arise anywhere in the gastrointestinal tract from the oesophagus to the anus. Historically considered a rare cancer type (1 and 3.5 new cases annually/100,000 people in Europe and the United States of America respectively), the incidence of GEP-NENs is increasing across all primary sites, stages and grades of disease, with a three-fold increase in cases observed over the last 40 years [2,3]. 

Investigation of the NET genetic landscape is characterised predominantly by small, heterogeneous studies driven by advances in next-generation sequencing and other high-throughput technologies. As a result, large numbers of genetic and epigenetic changes have been identified. However, despite the large amount of data, it has not yet been possible to identify key dysregulated pathways that drive carcinogenesis in NETs, a process that would be addressed by large tissue profiling studies that integrate DNA, RNA and methylation. It is clear that there is a paucity of genetic mutations in classical tumour suppressor and oncogenes, which has fueled interest in the role of epigenetics in both oncogenesis and tumour evolution in GEP-NETs. 

Epigenetics refers to a change in gene expression that does not involve a change to the underlying DNA sequence—a change in phenotype without a change in genotype [4]. Epigenetic dysregulation has been demonstrated in both the primary NET and the surrounding healthy parenchyma, and is believed to contribute to the development and progression of NETs [2]. Identifying and potentially modulating epigenetic dysregulation represents a viable treatment strategy, as unlike somatic mutations, epigenetic changes are potentially reversible, translating to a potentially more attractive ‘druggable’ target. 

This review explores the current understanding of the natural history of GEP-NETs, with specific reference to underlying epigenetic changes and relevance to tumorigenesis. We briefly summarise the main epigenetic pathways before exploring how these are altered or augmented in GEP-NETs. We look at novel biomarkers and potential avenues for therapeutic modulation utilising both pharmacological and theranostic approaches. Finally, we look at the potential difficulties and barriers to research in GEP-NETs and how these may be overcome.

## 2. Epigenetic Modifications in Cancer

Epigenetics refers to a change in phenotype without a corresponding alteration in the genotype [5]. There are a number of epigenetic mechanisms, including DNA methylation, histone modification and microRNAs (miRNAs), which are involved in post-translational modification. 

### 2.1. DNA Methylation

DNA promoter methylation is one of the most well-recognised epigenetic modifications in both normal and cancer cells. The methylation of cytosine to 5-methylcytosine is so integral to genomic homeostasis that this is often referred to as the ‘fifth base’ of the genome [6]. Methylation typically occurs within the CpG (or CpG dinucleotides) island via the action of DNA methyltransferase enzymes (DNMT) [7,8,9]. There are three main families of DNMT enzymes: DNMT1 maintains methylation, while DNMT3 consists of two related gene products (DNMT3a and DNMT3b) which are responsible for de novo methylation. The function of DNMT2 has not yet been fully elucidated. CpG islands are commonly found in the upstream promoter region of genes and are responsible for recruiting inhibitory protein complexes leading to transcriptional repression of the downstream gene. Methylation affects chromatin structure at high CpG-density sites, either by recruiting inhibitory protein complexes or by inhibiting the binding of transcription factors, resulting in transcriptional repression of downstream genes [10,11]. Conversely, in general, when promoter regions are hypomethylated, downstream genes are expressed [12]. DNA methylation also has an essential function in maintaining genome stability; if global hypomethylation occurs, the genome may become unstable [13]. Of particular note in tumorigenesis, global hypomethylation may co-exist with gene-specific hypermethylation, suggesting a degree of coordination in this dysregulation [13].

### 2.2. Histone Modifications

Histones are highly basic proteins that act as spools around which DNA intertwines with chromatin, forming tight compact structures known as nucleosomes. Histone remodelling is a key gene expression regulator and is primarily carried out by two enzymes, histone deacetylases (HDACs) and histone acetyltransferases (HATs), with inverse functions. HDACs catalyse the removal of acetyl groups, resulting in a closed chromatin structure and generally suppressing gene expression. HATs, in contrast, mediate the reversible acetylation of histones and non-histone proteins, resulting in a local “open chromatin” and allowing gene expression concomitantly through the binding of multiple transcription factors. Other key histone post-translational modifications include methylation, ubiquitination and phosphorylation, which similarly can increase or decrease the transcription of genes [10].

### 2.3. MicroRNAs

MicroRNAs (miRNAs) are highly conserved, single-stranded non-coding molecules of RNA that are approximately 21 to 25 nucleotides in length. They regulate protein expression by targeting mRNAs for degradation or transcriptional repression [14,15]. miRNA expression has been demonstrated to be dysregulated in a plethora of human cancers, including both hematopoietic and epithelial cancers, impacting proliferative signalling, apoptosis, metastasis and angiogenesis [16].

## 3. Pancreatic Neuroendocrine Tumours

Pancreatic NETs (pNETs) comprise 15–20% of all GEP-NETs [17]. They are a heterogeneous group of tumours with variable clinical behavior, depending on histologic features and disease staging. The overall 5-year survival is 60% for limited stage disease and 25% for metastatic disease [18]. Although most pNETs are non-functional (45–60%), functioning pNETs (e.g., insulinoma, gastrinoma, glucagonoma, VIPoma and somatostatinoma) display a plethora of hormone-related syndrome-types according to the different hormones produced by the exocrine cells of the pancreas. A small number of pNETs (10%) are associated with genetic syndromes; MEN1, NF1, tuberous sclerosis and von Hippel-Lindau, which impact epigenetic regulation, are discussed below. pNETs are the most well-characterised group in terms of molecular profiling, and of these, insulinomas have emerged as having a distinct epigenetic pattern, which has potential utility both in terms of diagnosis and therapy.

### 3.1. Promoter Hypermethylation in pNETs

***Ras-association domain gene family 1 (RASSF1)*** is a tumour suppressor gene which functions to arrest the cell cycle in G1 by a mechanism that leads to accumulation of cyclin D1 (Figure 1) [19]. Downregulation of this pathway has been reported in at least 37 different tumour types, including GEP-NETs, and this downregulation is almost universally secondary to promoter methylation [19,20]. The RASSF1 regulatory region contains two promoter regions, denoted as A and C, which encode for eight RASSF1 isoforms. Selective methylation leads to the production of only 2 different isoforms: RASSF1A and RASSF1C. RASSF1A regulates cellular proliferation, apoptosis and stabilisation of microtubules and is the most hypermethylated gene in pNETs (75–83%) [20,21,22,23,24]. In the largest study of DNA methylation to date, House and colleagues investigated promoter methylation of 11 candidate tumour suppressor genes in 48 well-differentiated pNETs [22]. Aberrant methylation was reported in 87% of tumours studied, with the most common gene being RASSF1A, which was methylated in 75% of cases. No methylation of RASSF1A was observed in the surrounding normal tissue, suggesting RASSF1A methylation is an important epigenetic change driving tumorigenesis [22]. The authors also report an association between a higher frequency of promoter methylation with tumours larger than 5 cm and those with either lymph node or hepatic metastases, again suggesting the role of methylation in carcinogenesis. RASSF1A promoter methylation is less frequently present in pancreatic adenocarcinoma, suggesting methylation of RASSF1A is more specific to producing a neuroendocrine pancreatic tumour phenotype [20]. 

Mapeli and colleagues demonstrated that RASSF1C expression was significantly higher in pNET compared to normal tissue (11.4x, *p* = 0.001), where RASSF1C is believed to play a role in the upregulation of the Wnt pathway through inhibition of β-catenin degradation, suggesting that RASSF1C may also have a pathogenic role in the development of pNETs [25]. Moreover, death domain-associated protein (DAXX) retains RASSF1C within the nucleus, releasing RASSF1C on DAXX degradation with DNA damage [26]. However, the overexpression of RASSF1C was not found to be associated with promoter methylation [25]. 

***Cyclin-dependent kinase inhibitor 2a/P16INK4a (CDKN2A)*** encodes the tumour suppressor protein p16, whose primary role is to regulate cell entry into the S-phase of the cell cycle. CDKN2A has been shown to be silenced in many tumour types through promoter methylation of CDKN2A/P16 locus [27]. Lubomierski and colleagues investigated the implication of inactivation of CDKN2A in 37 primary GEP-NETs and two NET cell lines [28]. They found CDKN2A methylation to be present in 57% of pNET samples studied as well as in the pNET cell line QGP1. House and colleagues identified hypermethylation of CDKN2A in 40% of 48 patients with pNETs who underwent tumour resection. This was not present in the adjacent normal tissue studied, and multivariate analysis indicated that hypermethylation of CDKN2A may be an independent negative predictor of patient survival following surgical resection, a feature that has been shown to associated with the presence of metastases and poor 5-year progression-free survival in other studies [22,24,29,30]. 

Differential methylation of the CDKN2A promoter has also been demonstrated in subtypes of pNETs. Two case series of gastrinomas reported CDKN2A promoter methylation in 58% of cases, independent of stage and prognosis, suggesting that this is an early occurrence in tumorigenesis [30,31]. In contrast, insulinomas showed low levels of CDKN2A promoter methylation (17%, N = 17) [32]. Whilst only limited conclusions can be reached due to the small sample size, these findings warrant further investigation. 

***Tissue inhibitor of metalloproteinase-3 (TIMP3)*** is a putative tumour suppressor gene which inhibits metalloproteinase, resulting in reduced cellular growth, cellular migration and invasion. In a series of 18 matched tumour and healthy donor tissue samples, Wild and colleagues identified TIMP3 promotor methylation in 44% of tissue tumour samples [33]. They found this epigenetic change to be tumour specific and found that presence of hypermethylation at this locus corresponded to a reduction, and in some cases complete loss of, TIMP3 expression results. This finding was mirrored by a study by Stefanoli and colleagues but not observed in two studies by Arnold and colleagues [23,29,34]. In the study by Wild, 79% of patients with metastatic disease had TIMP3 methylation, compared with 14% of patients without [33]. Interestingly, none of the insulinoma samples in the Wild study displayed TIMP3 hypermethylation, further informing the observation of differential methylation patterns in pNET subtypes [33].

***O-6-alkylguanine-DNA alkyltransferase (MGMT)*** is a DNA repair enzyme that leads to rapid reversal of alkylation of the O-6 position of guanine bases (O6MG), preventing the formation of unrepairable DNA cross-links (Figure 2) [35]. In the absence of repair, O6MG is erroneously paired with thymine, creating a pro-mutagenic O-6 MGMT mismatch [36,37]. MGMT also inhibits the binding of transcription factors and other cellular regulators to gene promoters [38]. In glioblastoma multiforme, MGMT promoter methylation status is predictive of response to alkylating agents, such as temozolomide [39]. This effect has also been demonstrated in GEP-NET; in a study of patients with well-differentiated, advanced NETs, a significantly longer median progression free survival was seen in patients with MGMT promoter methylation treated with an alkylating agent. This therefore shows utility as a therapeutic biomarker of response; MGMT hypermethylation has been observed in 17 to 50% of pNETs [22,23,40,41]. However, the literature is conflicted in part due to the small, retrospective nature of the published studies as well as the multiple techniques used to assess MGMT status, all of which have varying accuracy [42,43]. The role of MGMT status is currently being investigated as part of a phase II randomised study where patients will be randomised to an alkylating agent or oxaliplatin based on MGMT methylation status [44]. Furthermore, the ‘hypermethylator phenotype’, defined as the presence of more than three hypermethylated genes, was found to be predictive of harboring MGMT promoter hypermethylation, and in this scenario was associated with an adverse clinical picture, highlighting that MGMT may also have a role in prognostication [40].

**Insulin Growth Factor-2 gene** (***IGF2***) is a gene that is imprinted through hypermethylation. Loss of imprinting and overexpression of IGF2 has been described in pNETs [45]. Dejeux and colleagues investigated the methylation profile of IGF2 in 62 patients with either small bowel or pNETs, including 11 insulinomas. The results illustrated that hypermethylation of region 2 was specific to insulinomas [46]. The authors also observed a relationship between increasing malignancy and a decreasing degree of methylation in IGF2, suggesting the potential role of IGF2 methylation as a biomarker for staging and classification of pNETs.

### 3.2. Methylation of Other Loci in pNET

Widepsread CpG island promoter methylation, also referred to as CpG island methylator phenotype (CIMP), has been observed in a significant percentage of pNETs studied (83%) and is associated with the presence of metastases and worse prognosis [23].

House and colleagues reported promoter methylation of MLH1 in two studies associated with microsatellite instability [22,47]. This was observed by Mei and colleagues in a study of 55 insulinomas where they observed promoter hypermethylation of MLH1 in 36% of cases. This was associated with microsatellite instability in 31% of cases, and the presence of both biomarkers was noted to be associated with poor prognosis [48].

Long interspersed nucleotide element 1 (LINE1) and Arthrobacter luteus (ALU) homolog are non-coding genomic repetitive sequences, the methylation status of which can be used as a surrogate marker of global hypomethylation [49]. Hypomethylation of these regions compared with adjacent non-cancerous pancreatic tissue has been observed in 20–33% of cases and is associated with poor prognosis [34,50].

### 3.3. Histone and Chromatin Modification

The most frequently mutated genes in pNETs are MEN1, DAXX or alpha thalassemia/mental retardation syndrome X-linked (ATRX), occurring in 44%, 25% and 18% of pNETs, respectively [51]. Mutations of menin and DAXX/ATRX have not been shown to be mutually exclusive in the same tumour. MEN1 and DAXX/ATRX are part of chromatin modifying complexes. MEN1 is a tumour suppressor gene that encodes the transcription factor menin. Menin regulates the methylation of histone H3 at lysine residue 4 (H3K4me3) and recruits the nuclear complex mixed-lineage leukaemia members 1 and 2 (MLL1/2) [52]. MLL binds to the promoter regions of cyclin-dependent kinase inhibitors (CDKis) p18Ink4c (CDKN2C) and p27Kip1 (CDKN1B), maintaining the expression of these genes and inhibiting tumour formation [53]. With MLL loss or with disruption of this complex by menin, methylation levels are reduced, resulting in reduced CDKi expression and tumour growth [54]. MEN1 has also been observed to interact with other HDACs and histone methyltransferases, including SUV39H1, acting as either an activator or suppressor of gene transcriptional activity [55,56].

DAXX is a histone H3.3 chaperone which is guided by ATRX and is a nuclear protein of the SWI/ SNF complex of chromatin-remodeling genes. ATRX recruits DAXX, mediating DAXX-dependent H3.3 deposition at H3K9me3-enriched chromatin and telomeres where it mediates both chromatin remodelling and telomere length [57]. Loss-of-function mutations in DAXX and ATRX lead to an exaggerated DNA damage response, the alternative lengthening of telomeres (ALT) pathway, and genomic instability. Previous studies have illustrated 100% concordance in pNETs with DAXX or ATRX mutations and the ALT phenotype [51,58]. In addition, DAXX, along with p53, impacts methylation by directing DNMT1 to the promoter region of RASSF1A, as demonstrated by Zhang and colleagues in a leukaemia model [59]. ATRX/DAXX loss/mutations and ALT positivity are associated with a more aggressive tumour phenotype (larger tumours, grade stage), chromosomal instability, metastatic disease and survival, with the absence of ATRX/DAXX being an independent predictor of survival on multivariable analysis [60].

### 3.4. MicroRNA (miRNA)

Whilst miRNAs are one of the most abundant classes of gene-regulatory molecules, few studies have considered the role of miRNA in the pathogenesis of GEP-NET. Roldo and colleagues investigated global miRNA expression in the normal pancreas. pNETs (twelve insulinomas and 28 non-functioning endocrine tumours) and four acinocarcinomas were evaluated regarding the involvement of miRNA in malignant transformation and progression [61]. Results demonstrated increased expression of miRNA-103 and miRNA-107 and reduced expression of miRNA-155 in tumour tissue compared to normal pancreas tissue. miRNA-21 overexpression in pNETs was strongly associated with both a high Ki-67 proliferation index and the presence of liver metastases, both of which are recognised as poor prognostic factors. Furthermore, the author identified a panel of ten miRNAs (miRNA-125a, -99a, -99b, -125b-1, -342, -130a, -132, -129-2 and -125b-2) able to distinguish between endocrine and acinar tumours. These miRNAs also had a possible association with either normal endocrine differentiation or endocrine tumorigenesis.

In a similar study, Thorns and colleagues studied the miRNA profiles of 37 pNETS compared with non-neoplastic pancreas and micro-dissected islet cells [62]. The authors observed different miRNA signatures between pNET, pancreatic islets and total pancreas, with virtually no overlap. In particular, they observed that expression of miRNA-642 correlated with Ki67, and miRNA-210 correlated with metastatic disease, but not miRNA-21. Grolmusz and colleagues reanalysed the previously published miRNA data and then performed further validation in an independent dataset [63]. They observed two clusters of pNETs based upon miRNA expression signatures: cluster 1, which was less proliferative, and a more proliferative cluster 2. miRNA-21 was confirmed to be associated with Ki67 and the presence of metastatic disease and survival.

In a large study of 40 pNETs, three distinct subsets named miR-cluster-1, miR-cluster-2, and miR-cluster-3 were identified which differentially expressed 30 distinct miRNAs. miR-cluster-1 included MEN1-mutant tumours with moderate metastatic potential. miR-cluster-2 was enriched in metastasis-like primaries (MPL) with high metastatic potential, whilst miR-cluster-3 predominantly included insulinomas, none of which were associated with metastatic disease [64]. The authors observed that 21% of cluster 1 tumours also contained *DAXX/ATRX* alterations, again lending support to the concept that these mutations are late events in pNET pathogenesis driven by *MEN1* loss.

## 4. Gasterointestinal NETs

NETs of the gastrointestinal tract (GI-NETs) comprise over two third of GEP-NETs^30^. They are classically divided into foregut (oesophagus, stomach, duodenum), midgut (appendix, small bowel, caecum and ascending large bowel) and hindgut (distal large bowel, rectum). Epigenetic interest in GI-NETs has been, at least in part, fuelled by increased understanding of the role of methylation and CpG island methylator phenotype positivity (denoting a high degree of methylation across multiple CpG sites) in colorectal adenocarcinoma onset and progression. Research into histone modifications and miRNA in SI-NETs has been less informative.

### 4.1. Promoter Methylation

#### 4.1.1. RASSF1A

As seen in pNETs, GI-NETs are also frequently methylated in the RASSF1A promoter region. Pizzi and colleagues identified RASSF1A promoter methylation in 32% of 62 GI-NETs studied, although this was restricted to foregut tumours [21]. Of these tumours, cyclin D1 hyper-expression was found in 53% of cases and correlated with RASSF1A methylation.

In a second study of 33 midgut tumours, Zhang and colleagues found that RASSF1A and *CTNNB1* promoter region methylation were more frequent in metastatic lesions compared to primary tumours [65]. A majority of primary tumours (61%) exhibited RASSF1A promoter methylation, whereas 85% of matched metastatic lesions exhibited RASSF1A promoter methylation. In addition, methylation of the cadherin-associated protein (CTNNB1) promoter region was found to be associated with the development of metastatic disease. Study results demonstrated that 58% of metastatic GI-NETs exhibited CTNNB1 promoter methylation, which was absent in corresponding primary lesions [21]. Additionally, methylation of the CTNNB1 promoter region was not reported in surrounding healthy enterochromaffin cells, which is supportive of its role in oncogenesis and metastasis. There was no evidence of either RASSF1A or CTNNB1 methylation in the 6 appendiceal GI-NETs studied. Appendiceal NETs have a low metastatic potential and generally a more indolent course. This study suggests that methylation status of RASSF1A/CTNNB1 promoters may be involved in the development of foregut and midgut NETs rather than hindgut lesions. In a further study in 44 SI-NETs, promoter methylation was again observed for RASSF1A and CTNNB1 [66]. This study also investigated long interspersed nucleotide element 1 (LINE1) methylation and found an association with loss of chromosome 18. LINE1 and Alu are non-coding genomic repetitive sequences, the methylation status of which can be used as a surrogate marker of global hypomethylation [49]. Global hypomethylation has been shown to correlate with deletion of the short arm of chromosome 18, lymph node metastatic disease and reduced survival in pNETs [50]. Whilst previously thought to be a feature more typical of SI-NETs [50], Stricker and colleagues have identified global hypomethylation occurring in other types of GEP-NETs and have found that global hypomethylation occurs in 100% of the 15 pNETs studied [67].

Given the observed loss of chromosome 18 in SI-NETs, Edfelt and colleagues investigated the epigenetic regulation of TCEB3C as the only imprinted gene on this chromosome that encodes for elongin A3 [68]. The expression of elongin A3 was low in 77% of SI-NETs studies and was associated with the presence of only one copy of TCEB3C in 89% of cases, and methylation of the promoter region of this gene was observed in a subset of patients.

In a study of 101 SI-NETs, the gene SEMA3F, which encodes for semaphorin 3, was observed to be methylated in 50% of tumours and correlated with a higher Ki67 and tumour stage, an observation which warrants further investigation [69].

#### 4.1.2. Global Methylation Patterns

A number of high throughput studies have been undertaken to study global methylation patterns in SI-NETs. Verdugo and colleagues investigated DNA methylation in 20 SI-NETs and their corresponding metastases [70]. They reported that CpG islands were less methylated in metastases compared to primary tumours, thereby indicating the presence of global hypomethylation. Of interest, a loss of heterozygosity was observed at chromosome 18. Additionally, supporting previous findings, the promoter region of TCEB3C was observed to be methylated. This provides evidence to support the hypothesis that hypomethylation contributes to malignant behaviour through genomic instability and may represent a potential therapeutic strategy to reduce metastatic potential.

Karpathakis and colleagues conducted the largest genome wide study to date, integrating genomic, epigenomic, and transcriptomic analysis to analyse 91 SI-NETs [71]. The authors identified 3 distinct groups based on survival following resection. In terms of epimutations, 21 epigenetically dysregulated genes were identified at a recurrence rate of up to 85%, including CDX1 (86%), CELSR3 (84%), FBP1 (84%), and GIPR (74%). In a subsequent paper, the authors compared the methylation patterns in the identified genes between liver metastases and the primary SI-NETs [72]. Progressive aberrancy of methylation in a panel of 21 epimutated genes in SI-NET liver metastases compared to the primary SI-NET indicated that more advanced epigenetic dysregulation could be a driving factor towards metastatic disease. It is important, however, to consider the small number of cases involved in these studies, which limits the ability to achieve statistical significance.

Rahman and colleagues investigated the expression of DNMT1, -3A and 3B in 63 GEP-NETs [73]. They reported a significant increase in DNMT3A expression in poorly differentiated tumours compared to well-differentiated tumours (*p* > 0.01). The expression of DNMT1, 3A and 3B were all higher in advanced stage tumours and correlated with the proliferation index. Taken together, these findings suggest that DNMT may be a therapeutic target that warrants further investigation.

Arnold and colleagues investigated MSI and CIMP in 34 malignant poorly differentiated colorectal NET, as well as in 38 well-differentiated benign and malignant fore-/midgut-NETs [29]. Of the 34 colorectal NETs, 60% of the MSI-low and 68% of the MS-stable tumours were CIMP-positive. None of the foregut NETs were MSI-high. CIMP status did not affect survival. Of the methylated genes, p16, hMLH1 and TIMP3 were differentially methylated in colorectal NETs, where p16 was found to be an independent predictor of outcome.

### 4.2. Histone and Chromatin Modification

Studies looking into histone and chromatin modification in GI-NETs have been limited to small case-controlled studies of primarily SI-NETs, and more research is required in this area. The histones H1 (linker histone) family have a number of genomic roles including promoting structural stability, regulating gene expression and participating in chromatin-based DNA replication and repair. Warnebolt and colleagues showed that, of the 11 H1 homologous proteins described, the H1x subtype is significantly overexpressed in both SI-NETs and pNETs compared to surrounding normal tissue [74].

Methylation of core histones (including histone H3) regulate chromatin structure and gene expression. Histone H3 undergoes extensive post-translational modification and is known to have a number of epigenetic functions in various cancers and beyond (e.g., in drug addiction) [75]. Magerl and colleagues studied demethylation of histone H3 at lysine 4 (H3K4diMe), demethylating enzymes (LSD1) and H3K4 methylation (Ash2 complex) in a range of cancers, including NETs [76]. Strikingly, 100% of neuroendocrine carcinomas showed strong immunostaining for both LSD1 and Ash2. Furthermore, 93% showed demethylation of H3K4diMe, with only corresponding weak expression in matched normal tissue. Considering the emerging role of histone H3 in other cancers, this observation warrants further exploration.

### 4.3. MicroRNA

Research into MiRNAs in GI-NETs has thus far been relatively limited in comparison to other cancers. Studies have primarily focused on expression differences between primary tumours and metastatic deposits and have suggested a role in disease progression, potentially showing utility in detecting more aggressive variants. Ruebel and colleagues analysed 95 miRNAs in eight matched primary and metastatic ileal NET tumours [77]. They showed downregulation of miRNA-133a, -145, -146, -222, and -10b in all samples and upregulation of miRNA-183, -488, and -19a + b in six of eight metastatic tumours compared to the primary site tissue. Of particular interest, this study demonstrated changes to miRNA-133a expression, which has been implicated in the tumorigenesis and progression of a number of other digestive system cancers (e.g., oesophagus, gastric, pancreas, hepatic and colorectal) [78]. Other studies have consistently shown differential expression in SI-NETs, including upregulation of miRNA-96 and 183 and downregulation of miRNA-129-5p, 133a and 143-3p [79,80]. However further research and validation is necessary but is of significant interest.

The utility of miRNA as a biomarker for diagnoses has also been explored in GI-NETs. Hamfjord and colleagues showed dysregulation of miRNA expression in paired samples of seven colorectal adenocarcinomas and one colonic NET with normal mucosa [81]. The expression of 38 miRNAs was upregulated in the colonic NET, compared with only 6 miRNAs in the colorectal adenocarcinoma, suggesting a distinct expression pattern between these neoplasms. Despite this striking finding, excitement should be tempered as this study used normal mucosa as a comparator, not benign neuroendocrine tissue, and is limited to only a single case. However, several miRNAs identified, such as miRNA-96 and 182, have been noted in previous studies discussed earlier, making excellent candidates for further investigation. A further study by Arvidsson and colleagues in SI-NETs has shown that miRNA has potential as a prognostic marker correlating with proliferation index (Ki-67) and tumour progression (higher expression of miRNA-95 and 210, and lower expression of 378a-3p) [82]. Of particular significance was the identification of expression of miRNA-375 in over 90% of tumour samples. Expression of miRNA-375 is regulated by the transcription factor NeuroD1, an important regulator of enteroendocrine lineage, raising the possibility that this may regulate neuroendocrine transmutation to GI-NETs [83].

The potential utility of miRNAs as serum biomarkers in GI-NETs has been recently explored by Bowden and colleagues [84]. In a multistage study, they refined a 31-candidate miRNA panel in patients with SI-NETs and healthy controls. They identified that increased miRNA-21-5p and 22-3p and low levels of miRNA-150-5p were associated with the presence of metastatic SI-NETs, and higher concentrations correlated with shorter survival time.

## 5. Targeting Epigenetic Changes for Therapeutic Benefit

Epigenetic modifications play critical roles in tumorigenesis by altering the expression of oncogenes and tumour suppresser genes through various mechanisms, including DNA methylation, histone and chromatin remodeling and miRNA expression. These epigenetic modifications are reversible, opening an innovative area of drug development in neuroendocrine cancers. With increased understanding of the epigenetics in GEP-NETs, we review the potential for translation into meaningful treatments for patients.

### 5.1. DNA Methyltransferase Inhibitors

DNA methyltransferase inhibitors (DNMTIs) are one of the most successful classes of epigenetic drugs for the treatment of malignancies, such as myelomonocytic leukaemia (Figure 3). The prototype DNMTI is decitabine, a cytidine analogue (modified in position 5 of the pyrimidine ring). Decitabine is transported into cells by human concentrative nucleoside transporter-1, is converted into its active form by deoxycytidine kinase to 5-aza-2′-deoxycytidine-5′-triphosphate and is degraded by cytidine deaminase (CDA) [4]. The deoxyribose analogue is incorporated into DNA strands only in an S-phase dependent manner. The incorporated 5-azanucleoside (5-AZA-dC) disrupts the interaction between DNA and DNMTs and promotes proteasomal degradation of the DNMT. This prevents further methylation of cytosine residues, causing downstream inhibition of DNA methylation in daughter cells following division (Figure 4). Due to its pharmacological properties, at therapeutic concentrations decitabine is only active during the S-phase of the cell cycle, proffering preferential toxicity on rapidly dividing cancer cells. Normal cells are relatively unaffected, with the notable exception of haematopoietic cells. In clinical trials, achieving steady state blood therapeutic concentrations of decitabine has been problematic due to its pharmacokinetic profile. Haematological toxicities, such as neutropenia and thrombocytopenia, are frequently dose-limiting; these issues are circumvented with the introduction of second-generation demethylating agents [85].

Alexander and colleagues showed the effects of 5-aza-dC on three carcinoid cell lines—human ileal (CDNT2.5), pulmonary (H727), and pancreatic (BON-1) [86]. A dose-dependent reduction in cell proliferation and growth was observed in all three lines with a significant reduction of neuroendocrine markers (e.g., chromogranin A) observed in the ileal line. This supports a potential role for DNMTIs in the control of GEP-NETs and the reduction of biogenic amine production. However, decitabine is restricted by rapid systemic deamination and hydrolysis, meaning multiple daily doses are required intravenously to reach a steady state within the therapeutic window. Second-generation DNMTIs, including guadecitabine (which has a dinucleotide structure comprising decitabine and deoxyguanosine), are resistant to cytidine deaminase and result in improved in vivo exposure times than intravenous decitabine [85]. Another novel agent, ASTX727, combines cedazuridine, a cytidine deaminase inhibitor, with decitabine, increasing oral bioavailability [87,88]. It is hoped that these newer DNMTIs will be less prone to degradation and thus more clinically useful. Despite a number of trials exploring the use of second generation DNMTIs in a range of solid cancers, no trials are looking at use in GEP-NETs, although there are trials ongoing in related cancer such as paraganglioma (NCT03165721).

### 5.2. Histone Deacetylation Inhibitors

Histone deacetylation inhibitors (HDACIs) have been tested in GEP-NETs with promising results, although the toxicity profile observed is similar to that observed with DNMTIs with problematic myelosuppression limiting clinical use [89].

One target of HDACIs is the p21 tumour suppressor gene, where upregulation leads to the blockade of the cyclin/CDK complex, leading to cell cycle arrest [90]. HDACIs also upregulated the intrinsic and extrinsic apoptosis pathways through the induction of proapoptotic genes Bmf, Bim, TRAIL and DR5 [91,92]. HDACIs may also decrease the expression of the vascular endothelial growth factor receptor (VEGFR), thereby reducing tumour angiogenesis. Furthermore, HIF-1α, a pro-angiogenic transcription factor, is hyperacetylated by HDACIs, resulting in its degradation [93]. The above evidence indicates the important role epigenetics has to play in controlling cellular proliferation and apoptosis and provides another potential avenue for further trials studying HDACIs as a potential therapy for GEP-NETs.

Baradari and colleagues investigated the antineoplastic effects of three different HDACIs, (trichostatin A (TSA), sodium butyrate (NaB), and MS-275, on the growth and apoptosis of GEP-NET cell lines CM and BON-1 [94]. All three HDACIs caused a dose-dependent inhibition of proliferation in both cell lines by inducing apoptosis. The efficacy of HDACIs on different cell lines highlights potential for efficacy; however, further exploratory work is needed.

Shah and colleagues explored the use of depsipeptide, an HDACI, in 15 patients with metastatic NETs [87]. Unfortunately, the project was prematurely terminated due to unexpected cardiotoxicity. This has been observed with other HDACIs, such as trichostatin, suggesting a class effect which may limit the clinical development of these drugs [95].

Kunnimalaiyaan and colleagues found that the activation of the Notch-1 gene significantly reduces tumour growth in vitro [96]. They hypothesised that the identification of compounds that activate the Notch-1 pathway in NETs could be a potential strategy to treat patients with NETs. A commonly used anti-epileptic drug, valproic acid (VPA), is known to be a potent HDACI [97]. Mohammed and colleagues studied the use of VPA for the treatment of NETs in eight patients [98]. Despite identifying upregulation of Notch-1 signaling, which increased up to 40 times in some patients, no overall tumour regression was seen in any patient, suggesting augmentation of this pathway may have only limited effects. However, due to the relative safety of VPA and the small sample size, further investigation should be performed.

## 6. Epigenetic Modification of the Somatostatin Receptor

NETs are characterised by the presence of somatostatin receptors (SSTRs) on cell surfaces, typically including subtypes SSTR2, and to a lesser extent SSTR5 [99]. Stable analogues (e.g. octreotide, lanreotide and pasireotide) of somatostatin (SSA) exert an anti-proliferative effect on GEP-NETs, directly inhibiting cell growth and downstream signaling pathways following binding to its cognate receptor [100,101]. Utilising the concept of theranostics, the efficacy of SSA has been further developed into peptide receptor radionuclide treatment (PRRT). This exploits the presence of SSTRs on the tumour surface, as indicated by functional PET imaging using radiolabelled SSAs (e.g., (^68^Ga)DOTA (0)-Phe (1)-Tyr (3)-octreotide (DOTATOC) or –DOTA-(Tyr^3^)-octreotate (DOTATATE)) and consists of radiolabeled SSAs with beta-emitting radioisotopes, either (^90^Y) or (^177^Lu), that bind specifically to SSTR on the tumour surface, becoming internalised and delivering tumoricidal radiation. The NETTER-1 trial reported a significant improvement in progression-free survival and overall survival in those patients with lower grade small bowel NETs receiving ^177^Lu-DOTA^0^-Tyr^3^–octreotate ((^177^Lu)DOTATATE) compared to single agent SSAs; this is approved in the EU for the management of grade 1 and 2 GEP-NETs [102]. However, a significant number of patients are unsuitable for (^177^Lu)DOTATATE due to limited SSTR2 expression. Exploring the role of epigenetics in the expression of SSTR2, Torrisani and colleagues identified a region in the promoter sequence of SSTR2 that undergoes promoter methylation, such that in vitro treatment with a demethylating agent, in this case decitabine, resulted in re-expression of SSTR2 in high-grade cell lines [103]. These findings were extended when Taelman and colleagues illustrated that treatment of NET cell lines both in vitro and in vivo with decitabine not only resulted in re-expression of SSTR2 but, importantly, also resulted in enhanced uptake of radiolabeled SSAs, showing functionality in receptor re-expression, a concept that we have independently corroborated [104]. Extending this concept forward, we propose a clinical trial in which patients receive ASTX727 to enhance SSTR2 expression, followed by PRRT, in which (^68^Ga)DOTA-PET is used to assess SSTR2 re-expression. Utilizing epigenetic modifiers to allow the re-expression of a receptor of interest has the potential to improve the clinical outcome in a significant number of GEP-NET patients who would otherwise not be suitable for PRRT.

## 7. Research Difficulties in GEP-NETs

Despite recent advances in both genetic and epigenetic research in GEP-NETs, significant barriers remain to effective research in this area. Current studies have been fragmented, caused by the relatively low incidence of GEP-NETs, limited availability of neoplastic tissue for integrated analysis and inadequate animal models to recapitulate the disease. Typical studies in GEP-NETs involve fewer than 100 disease cases and are further confounded by significant disease heterogenicity.

Disease models which fully recapitulate GEP-NETs are indispensable for the translation of pre-clinical programs to the clinic; however, attempts to establish patient-derived xenografts have been unsuccessful and currently available cell lines (such as GOT1 and BON-1) and mouse models more closely resemble poorly differentiated neuroendocrine carcinoma rather than well-differentiated NETs. The developing field of organoid research may potentially offer a new platform for GEP-NET research, with the potential to give valuable insight into both aetiology and progression.

The limited availability of preclinical models and human tissue of GEP-NETs and the performance of non-selective translational research has negatively impacted research, hindering the optimal development of treatments [105,106]. In the future, it will be essential to improve preclinical models and combine clinical and translational research methods, though recent utilisation of international collaboration in clinical trials has been encouraging and should be continued.

Ongoing projects, such as the National Institute for Health Epigenomics program, are generating new research tools, datasets and testing infrastructure to accelerate our understanding and have particular relevance for diseases such as GEP-NETs with significant epigenetic pathophysiology [107].

## 8. Conclusions

Cancer research has traditionally focused on genetic alterations which underpin tumorigenesis, leading to the defined hallmarks of cancer. However, in neoplasms with few somatic mutations, such as GEP-NETs, the increasing importance of epigenetic changes, such as methylation patterns, histone and chromatin remodeling alterations and miRNA, has been identified. Despite the limitations of small heterogeneous studies, the high frequency of mutations in genes encoding for epigenetic modifiers suggest a fundamental role in both the tumorigenesis and metastatic progression of GEP-NETs. Further dissection of the GEP-NET epigenome will be crucial for characterisation of specific epigenetic subtypes. Currently, the armamentarium for the management of GEP-NETs does not deliver personalized oncology, which has transformed the management of other cancers. The characterisation of specific epigenetic GEP-NETs subtypes will hopefully lead to better patient stratification for clinical trials and, ultimately, more effective therapies.

## Figures and Tables

**Figure 1 cancers-13-04181-f001:**
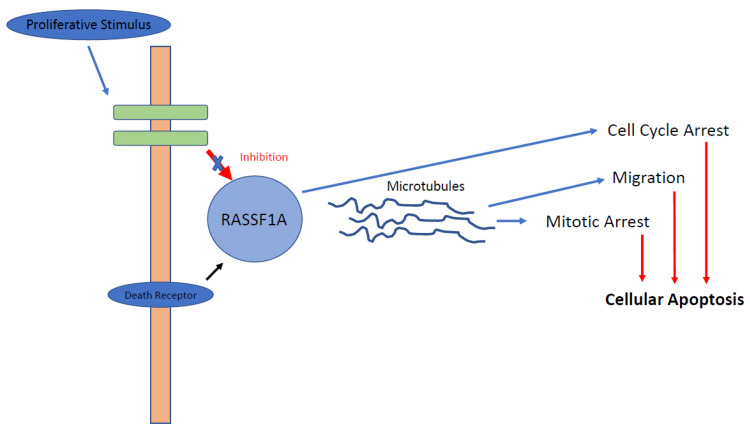
RASSF1A pathway. RASSF1A is affected positively by activation of the death receptor and affected negatively by proliferative stimuli, as demonstrated in GEP-NETs. Following activation of RASSF1A, binding to microtubule-binding proteins occurs, causing reduction in cellular migration, cell cycle arrest and, ultimately, apoptosis.

**Figure 2 cancers-13-04181-f002:**
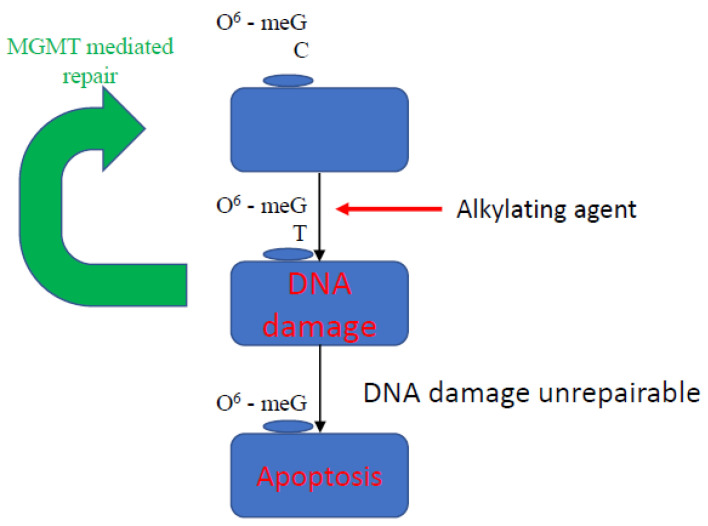
Mechanism of action of MGMT. O6-meG is removed by MGMT from 06-alkylguanine DNA. MGMT is inactivated and subject to ubiquitin degradation once the methyl group from the 06-meG is received. The 06-meGT is recognised by the mismatch repair pathway; subsequently, DNA double strands remain broken, and the cell undergoes apoptosis.

**Figure 3 cancers-13-04181-f003:**
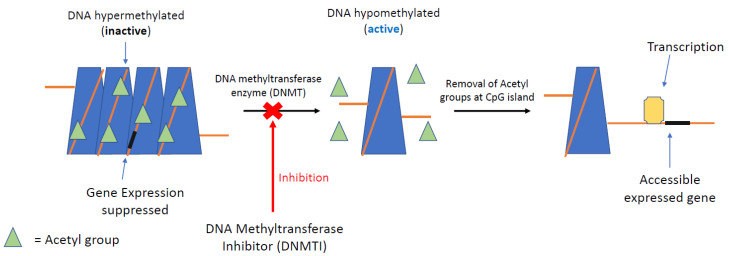
DNMTI mechanism of epigenetic regulation. DNA methylation at CpG islands (promoter) silences gene expression downstream. Removal of acetyl groups by DNA methyltransferase can re-establish gene expression, a process seen very commonly in GEP-NETs. Inhibition of DNA methyltransferase by selected inhibitors (DNMTI) can re-establish the gene, silencing suppressing transcription of the cancer genome.

**Figure 4 cancers-13-04181-f004:**
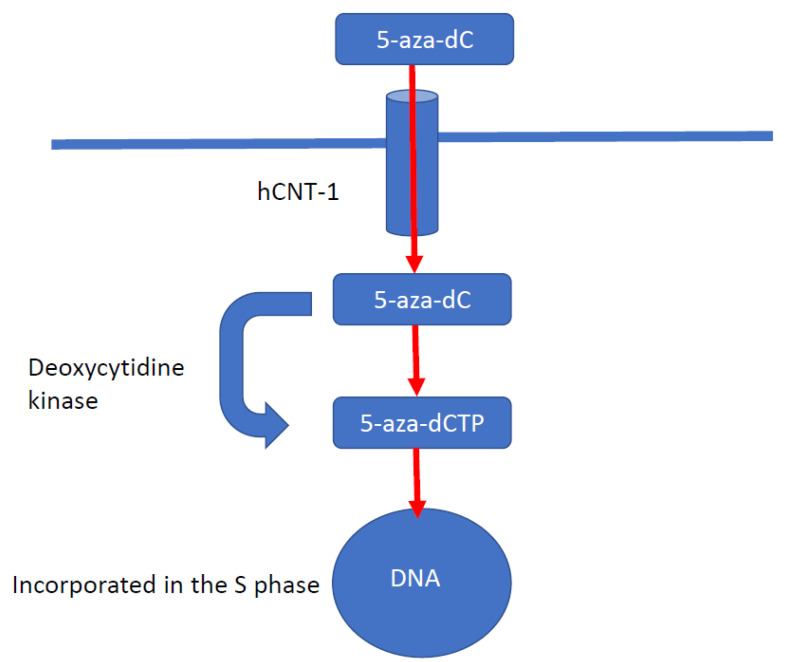
Mechanism of action of decitabine. Decitabine is a cytidine analogue modified in position 5 of the pyrimidine ring. Decitabine is transported into cells by human concentrative nucleoside transporter-1 (hCNT-1). Decitabine is converted by deoxycytidine kinase into its active form, 5-aza-2′-deoxycytidine-5′-triphosphate. Decitabine is a deoxyribose analogue and is incorporated only into DNA strands during the S phase of the cell cycle. The incorporated 5-azanucleoside disrupts the interaction between DNA and DNMTs and promotes proteosomal degradation of DNMT. This causes downstream inhibition of DNA methylation in subsequent daughter cells.

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
