# Peer review of "Exploring the Epigenome in Gastroenteropancreatic Neuroendocrine Neoplasias"

_cancers, 2021, doi:10.3390/cancers13164181_

Round 1

Reviewer 1 Report

This is an interesting and thorough review of published data on epigenetic mechanisms acting in neuroendocrine tumours (NET).

This review is timely and well written.

Only few items need to be addressed:

  1. In the introductory paragraph, it should be stated that the focus of the Review is the Neuroendocrine Tumours (NET), the well differentiated - and major – fraction of the neuroendocrine neoplasm (NEN) family. This to avoid potential misunderstanding that may arise along the paper, see for instance line 382 where “neuroendocrine carcinomas” are quoted, understandably referring to the poorly differentiated form of NEN, defined as neuroendocrine carcinoma (NEC).
  2. At line 29 it is stated that NET “…share common origin form embryonal gut…”, this is an obscure and strong statement unfortunately not supported by evidence – perhaps a more conservative statement should be used instead, given the paucity of evidence about the “origin” of NET; alternatively NET may just be defined as composed of tumour cell with a NE phenotype…
  3. At line 314 the word “that” is repeated twice.

Author Response

Thank you

We have corrected the manuscript as corrected. All changes have been saved as track changes in the document

Reviewer 2 Report

The objective of the article is clear
  The title is informative and relevant
  The references are: relevant and recent
Introduction / background
  It is clear what is already known on this subject
  The research question is clearly delineated
Methods
The variables are defined and measured appropriately
  The study methods are valid and reliable
Results
  The data are presented appropriately
Relevant and clearly presented tables and figures
Discussion and Conclusions
  The conclusions fit the objectives of the study
  Conclusions are supported by references or findings
Overall
The study design was appropriate to meet the purpose
The article is consistent in itself but added little to what is already known on this topic

Author Response

Thank you for your comments

This is a review article and the comments regarding SSTR are novel and not published elsewhere

Reviewer 3 Report

A well, extensive review well conducted

Author Response

Thank you for your kind words